

# Fasudil may alleviate alcohol-induced astrocyte damage by modifying lipid metabolism, as determined by metabonomics analysis

Huiying Zhao[1,*], Xintong Li[2,*], Yongqi Zheng[3], Xiaofeng Zhu[4], Xunzhong Qi[5], Xinyan Huang[6], Shunjie Bai[7], Chengji Wu[5] and Guangtao Sun[5]

[1] Department of Neurology, Jiamusi University, Jiamusi, Heilongjiang, China
[2] Department of Neurology, The Third Affiliated Hospital of Shenzhen University, Shenzhen, Guangdong, China
[3] Department of Internal Medicine, Yichun Forestry Administration Central Hospital, Yichun, Heilongjiang, China
[4] Department of Neurology, Mudanjiang Medical College, Mudanjiang, Heilongjiang, China
[5] Department of Neurology, The First Affiliated Hospital of Jiamusi University, Jiamusi, Heilongjiang, China
[6] Department of Neurology, The Second Affiliated Hospital of Jiamusi University, Jiamusi, Heilongjiang, China
[7] Department of Laboratory Medicine, The First Affiliated Hospital of Chongqing Medical University, Chongqing, Chongqing, China
* These authors contributed equally to this work.

Corresponding authors
Xiaofeng Zhu,
zhuxiaofeng_mdj@163.com
Xunzhong Qi, 617han@163.com

## ABSTRACT

Alcohol dependence is a chronic, relapsing encephalopathy characterized by compulsive craving for alcohol, loss of control over alcohol use, and the presence of negative emotions and physical discomfort when alcohol is unavailable. Harmful use of alcohol is one of the greatest risk factors for death, illness, and disability. Rho kinase inhibitors have neuroprotective effects. This study used metabonomics analysis to assess untreated astrocytes, astrocytes exposed to 75 mmol/L of alcohol, and astrocytes exposed to 75 mmol/L of alcohol and treated with 15 μg/mL fasudil for 24 h. One of the clearest differences between the alcohol-exposed and fasudil-treated alcohol-exposed groups was the abundance of lipids and lipid-like molecules, although glycerophospholipid metabolism was comparable in both groups. Our findings show that fasudil may alleviate alcohol-induced astrocyte damage by modifying lipid metabolism, providing a new approach for preventing and treating alcohol dependence.

## INTRODUCTION

Alcohol dependence (AD) is a disease involving excessive drinking in which individuals cannot control their own behavior and experience negative physical effects. AD has become a public health problem within all sectors of society. Alcohol has a wide range of

effects on the central nervous system (CNS), such as neurocytotoxicity, oxidative stress, and inflammation.

The role of astrocytes in AD is also receiving increasing attention. In patients with alcoholism, hippocampal glial cells were decreased by 37%, and astrocytes were decreased by 42% (*Korbo, 1999*). The response of astrocytes to alcohol exposure is not limited to changes in their numbers, morphology, and development; exposure can affect the many roles that astrocytes perform in the nervous system. These include regulation of neuroinflammatory processes, calcium signaling, balance of excitatory and inhibitory neurotransmission, water balance, cell volume regulation, and dopamine-dependent behavioral processes in the brain's reward circuits (*Adermark & Bowers, 2016*). In addition, acute or long-term exposure of astrocytes to alcohol may significantly alter connections between brain regions by interfering with myelin sheath maintenance (*Adermark & Bowers, 2016*). A comparison between alcohol-dependent and control individuals showed that the most differentially expressed genes, including genes that encode proteins and non-coding genes, could be detected in astrocytes, oligodendrocytes, and microglial cells (*Brenner et al., 2020*). Rho-associated coiled-coil containing kinases (ROCKs) include ROCK1 and ROCK2. Since their discovery, ROCKs have been extensively studied, revealing their multiple functions in the processes of cell contraction, migration, apoptosis, survival, and proliferation. Abnormal activation of the ROCK pathway can be observed in various CNS diseases (*Mueller, Mack & Teusch, 2005*), and favorable effects on many CNS diseases can also be observed with the application of ROCK inhibitors (*Bito et al., 2000*; *Satoh et al., 2008*; *Huentelman et al., 2009*; *Song et al., 2013*). In particular, inhibition of ROCK2 has been clearly shown to improve ethanol-mediated cognitive dysfunction in rats. ROCK2 can also reduce pathological damage in the hippocampus, neuronal death, inflammation, and oxidative stress (*Li et al., 2020a*). Fasudil, however, is the only ROCK inhibitor approved for human use to inhibit neurodegeneration. Therefore, fasudil has wide prospects for the prevention and treatment of neurological diseases in the future.

Currently, little is known about the effect of fasudil on small-molecule metabolites involved in alcohol-induced astrocyte damage. AD is a widespread disease, and finding an effective drug that can be developed and used is an inevitable trend and goal to reduce the influence of AD on individuals and society. On the premise that inhibition of the ROCK pathway has a protective effect in alcohol-induced astrocytes, 15 µg/mL of fasudil (as described in Metabolic Brain Disease; *Gao et al., 2019*) was used to treat astrocytes with alcohol-induced injury (75 mmol/L) for 24 h (*Li et al., 2018*). The concentration of alcohol was chosen because a blood alcohol concentration of 75 mM (*Adachi et al., 1991*) was found in 25 individuals with high alcohol intake, and this concentration was recommended for *in vitro* studies (*Deitrich & Harris, 1996*). This study explores the specific mechanism of ROCK inhibitors in the protection of alcohol-induced injury in astrocytes and provides new targets and experimental data for the pathogenesis, prevention, and treatment of AD.

## MATERIALS AND METHODS

### Materials

CTX TNA2 cells were purchased from American Type Culture Collection, USA. Dulbecco's Modified Eagle Medium was purchased from ShenGon Biotech, China. Fetal bovine serum was obtained from Gibco, Billings, MT, USA. Fasudil was obtained from Selleck, China. Trypsin-EDTA solution and phosphate-buffered saline (PBS) were obtained from Beyotime, Jiangsu, China. Ethanol was obtained from Tianjin Kaitong Chemical Reagent Company, Tianjin, China. TRIzol, chromatographic methanol, chromatographic pure acetonitrile, chromatographic pure formic acid, chromatographic pure water, and chromatographic pure propanol were purchased from Thermo Fisher Scientific, Waltham, MA, USA. The Q-Exactive HF-X mass spectrometer and Vanquish Horizon system were purchased from Thermo Fisher Scientific. BEH C18 columns were obtained from Waters Corporation, USA. The Jxdc-20 Nitrogen Purging Instrument was purchased from Shanghai Jingxin Industrial Development Company, Shanghai, China. The Lng-t88 Table Type Fast Centrifugal Concentrator was purchased from Taicang Huamei Biochemical Instrument Factory, Taicang, China. The Wonbio-96c High-throughput Tissue Crusher was obtained from Shanghai Wanbo Biotechnology Company, Shanghai, China. The Sbl-10td ultrasonic cleaning machine (300W-10L) was purchased from Ningbo Xinzhi Biotechnology Company, Ningbo, China. The 5430R high-speed refrigerated centrifuge was obtained from Eppendorf, Hamburg, Germany, and the NewClassic MF MS105DU electronic balance was purchased from Mettler, Greifensee, Switzerland.

### Cell groups and treatment conditions

Cells were divided into three groups (six culture dishes each): control, alcohol, and fasudil (fasudil + alcohol). CTX TNA2 astrocytes were inoculated into 75-cm$^2$ culture dishes at a density of $5 \times 105$ cells in 8 mL of complete cell culture medium (DMEM-H + 10% fetal bovine serum). When the cells reached 60% confluency, the growth medium was replaced with normal complete cell culture medium (control and alcohol groups) or medium containing 15 μg/mL fasudil (fasudil group). After culturing for 24 h, the medium was removed and replaced with normal complete cell culture medium (control group) or medium containing 75 mmol/L of alcohol (alcohol and fasudil groups). After an additional 24 h of culture, the culture medium was discarded, adherent cells were washed three times with 1 mL of PBS, and 2 mL of trypsin-EDTA solution was added to digest the cells. Subsequently, 4 mL of complete cell culture medium was added to terminate digestion, and the resulting cell suspensions were collected in 5-mL centrifuge tubes. After centrifugation at approximately $125 \times g$ for 5 min, the supernatant was discarded and cells were washed by resuspension in 1 mL of PBS and centrifugation at approximately $125 \times g$ for 5 min. Finally, the supernatant was discarded, and the resulting cell pellets were frozen in liquid nitrogen for 0.5 h before storage at −80 °C for later use.

## Metabonomics analysis

### Sample preparation

After thawing, 50 mg of each sample was added to 400 μL of a 1:1 mixture of acetonitrile and methanol, and the sample was fully mixed by vortexing for 30 s. Next, low-temperature ultrasonic extraction was performed for 30 min (5 °C, 40 kHz). The sample was then cooled at −20°C for 0.5 h, followed by centrifugation at 4 °C for 15 min at 125 × g. Next, the supernatant was discarded, the pellet was dried with nitrogen, and 120 μL of the extraction solution prepared above was added and mixed thoroughly to redissolve the pellet. Subsequently, another round of low-temperature ultrasonic extraction and centrifugation was performed at 4 °C for 5 min at 125 × g. The resulting supernatants were then stored in fresh tubes for later use.

### Quality control (QC) samples

Equal volumes of metabolites of all samples were mixed to prepare QC samples. During instrumental analysis, one QC sample was included with every 10 samples to investigate the repeatability of the analysis process.

## Liquid chromatography-mass spectrometry analysis

To ensure the stability and repeatability of the experiment, a QC sample was prepared by combining equal volumes of all samples and mixing thoroughly. Samples were analyzed on a UPLC-TripleToF mass spectrometer with a 0.4-mL/min flow rate and 40 °C column temperature. Mobile phase A consisted of water + 0.1% formic acid, while mobile phase B consisted of 49.95% acetonitrile + 49.95% ethylpropanol + 0.1% formic acid. Gradient elution was performed as follows: 0–3 min, 95–80% A and 5–20% B; 3–9 min, 80–5% A and 20–95% B; 9–13 min, 5% A and 95% B; 13.0–13.1 min, 5–95% A and 95–5% B; and 13.1–16 min, 95% A and 5% B. Samples were analyzed in positive and negative ion modes using the following parameters: 5.0-kV positive ion spray voltage, 5.0-kV negative ion spray voltage, 80-V cluster removal voltage, 500 °C ion heating temperature, 50 psi auxiliary heating gas, 50 psi spray gas, 30 psi curtain gas, and 20–60 V cycle collision energy.

## Metabonomics data analysis

The original liquid chromatography-mass spectrometry data were imported into Progenesis QI (V1.0) metabolomics processing software (Waters Corporation, Milford, MA, USA) for baseline filtering, peak identification, integration, retention time correction, and peak alignment. Finally, a data matrix containing the retention time, mass/charge ratio, and peak intensity was obtained. In the data matrix, the 80% rule was used to remove missing values; that is, variables with a non-zero value greater than 80% in at least one group of samples were retained, and then the missing value was added (the minimum value in the original matrix was used to replace the missing value). To reduce error caused by sample preparation and instrument instability, the response intensity of essential spectral peaks of samples was normalized using the summation normalization method, and the normalized data matrix was obtained. Additionally, variables with a relative

standard deviation >30% of that of the QC samples were excluded. The resulting data matrix was then matched with the Human Metabolome Database (HMDB, 2021; https://hmdb.ca) and Metlin database (Scripps Institute, 2017; https://metlin.scripps.edu) to identify metabolites. Preprocessed data were analyzed using the Majorbio platform (https://cloud.majorbio.com/). R software (version 1.6.2; *R Core Team, 2003*) was used to perform principal component analysis (PCA) and orthogonal projections to latent structures discriminant analysis (OPLS-DA) with seven-fold cross validation to avoid overfitting. Metabolites with variable importance in the projection (VIP) values > 1 and *P* values < 0.05 were selected as significant metabolites. Metabolic pathway annotation was carried out using the Kyoto Encyclopedia of Genes and Genomes (KEGG), and pathway enrichment analysis was carried out using Python software (version 2.7, https://www.python.org). The biological pathway most relevant to each experimental condition was determined using Fisher's exact test.

## RESULTS

### Effects of alcohol on astrocyte metabolomics

#### PCA analysis of samples from the alcohol and control groups

PCA analysis of the differences between groups showed that the alcohol and control groups tended to separate under anionic and cationic modes, but the relative concentration within each group was poor. Thus, PLS-DA supervisory analysis was needed to maximize differences between the two groups.

#### PLS-DA analysis of samples from the alcohol and control groups

PLS-DA is a statistical method used to perform supervised discriminant analysis. Control and alcohol group data were divided into two categories based on collection under the anionic mode or cationic mode (Fig. 1). In the cationic mode, the R2Y value was 0.752 and the Q2 value was 0.52. In the anionic mode, the R2Y value was 0.805 and the Q2 value was 0.333, indicating that PLS-DA showed no overfitting in the experiment, demonstrating the reliability of the data. Regardless of the mode, the PLS-DA score chart showed good inter-group differences and intra-group aggregation, indicating that the model had good explanatory and predictive performance.

#### Statistical analysis of differential metabolites

Twenty-eight differential metabolites were identified by comparing the control and alcohol groups ($p < 0.05$, VIP < 1; Fig. 2). Among them, six were upregulated and 22 were downregulated following exposure to alcohol. Comparison of differential metabolites with the HMDB database identified seven primary categories: lipids and lipid-like molecules (31.58%), such as palmitoyl glucuronide; Organoheterocyclic compounds (26.32%), such as L-nicotine; Organic acids and derivatives (10.53%), such as 3-amino-2-piperidone; Organic nitrogen compounds (10.53%), such as eicosapentaenoyl ethanolamide; Phenylpropanoids and polyketides (10.53%), such as 6″-malonylgenistin; Organic oxygen

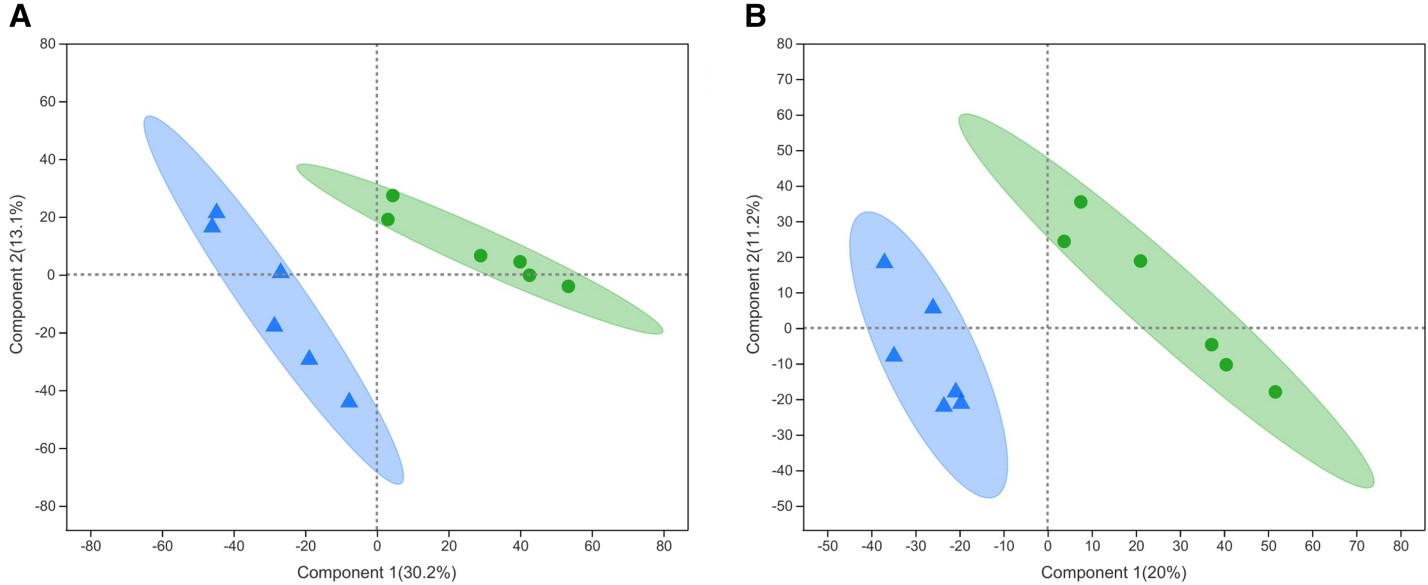

**Figure 1** **Projections to latent structures discriminant analysis (PLS-DA) score plots of astrocytes treated with different media.** Control: DMEM-H + 10% FBS, alcohol: normal medium containing 75 mmol/L alcohol. (A) PLS-DA model in the cationic mode. (B) PLS-DA model in the anionic mode. Δalcohol, Ocontrol.                

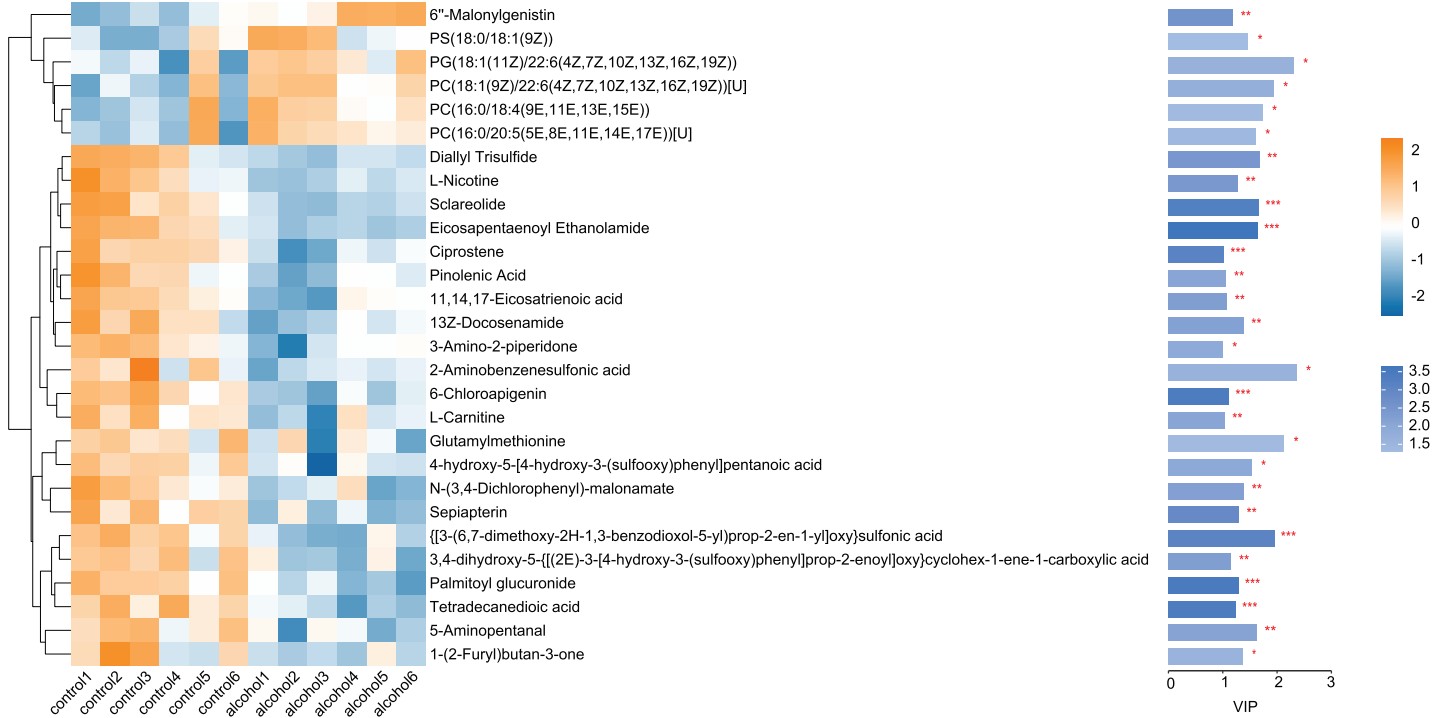

**Figure 2** **Heatmap of differential metabolites between the alcohol and control groups.** Yellow indicates a greater between-group difference, and blue indicates a lower between-group difference ($^*p < 0.05$, $^{**}p < 0.01$, or $^{***}p < 0.001$ *vs* control).

compounds (5.26%), such as 5-aminopentanal; and organosulfur compounds (5.26%), such as diallyl trisulfide (Fig. 3). Lipids and lipid-like molecules accounted for the highest

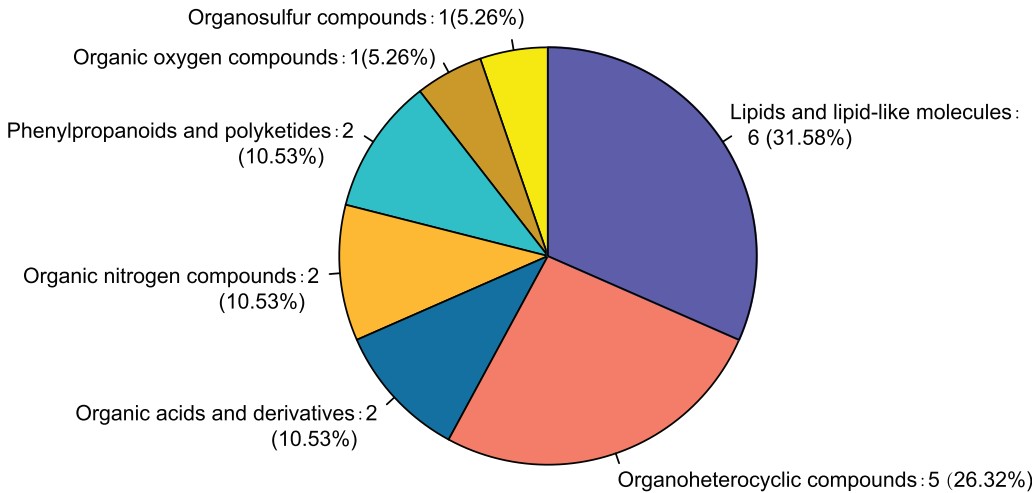

**Figure 3 Classification of differential metabolites and their proportion between the control and alcohol groups.**

proportion of differential metabolites, suggesting that lipid metabolism in astrocytes was abnormal after exposure to alcohol.

KEGG pathway analysis was performed to evaluate pathways enriched for differential metabolites between the control and alcohol groups. Thirteen pathways were identified, of which "Nicotine addiction" ($p = 0.0119$), "Thermogenesis" ($p = 0.0006$), "Systemic lupus erythematosus" ($p = 0.0051$), "Leishmaniasis" ($p = 0.0102$), "Amoebiasis" ($p = 0.022$), and "Bile secretion" ($p = 0.0108$) were the most significant. In addition, "Folate biosynthesis" ($p = 0.0935$), "Pentose and glucuronate interconversions" ($p = 0.0904$), "Lysine degradation" ($p = 0.0888$), "Biosynthesis of unsaturated fatty acids" ($p = 0.1199$), "Glycerophospholipid metabolism" ($p = 0.0856$), "Glycine, serine, and threonine metabolism" ($p = 0.0824$), and "Metabolism of xenobiotics by cytochrome P450" ($p = 0.1895$) were enriched (Fig. 4). Both the temperature-control and nicotine addiction pathways contain L-nicotine (VIP = 1.2742, $p = 0.0036$), which was significantly reduced in the alcohol group. This finding suggests that L-nicotine may be involved in the regulation of alcohol metabolism by astrocytes.

## Metabolomics analysis of the alcohol and fasudil groups
### *PCA analysis of samples from the alcohol and fasudil groups*
PCA was used to analyze differential metabolites between the alcohol and fasudil groups. Under the anionic and cationic modes, the alcohol and fasudil groups showed a tendency to separate, but the relative concentration within each group was poor. Thus, PLS-DA supervisory analysis was needed to maximize differences between the two groups.

### *PLS-DA analysis of samples from the alcohol and fasudil groups*
Metabolites from the control and alcohol groups were clearly separated under the anionic and cationic modes (Fig. 5). In the cationic mode, the R2Y value was 0.76 and the Q2 value was 0.268. In the anionic mode, the R2Y value was 0.235 and the Q2 value was 0.29. These

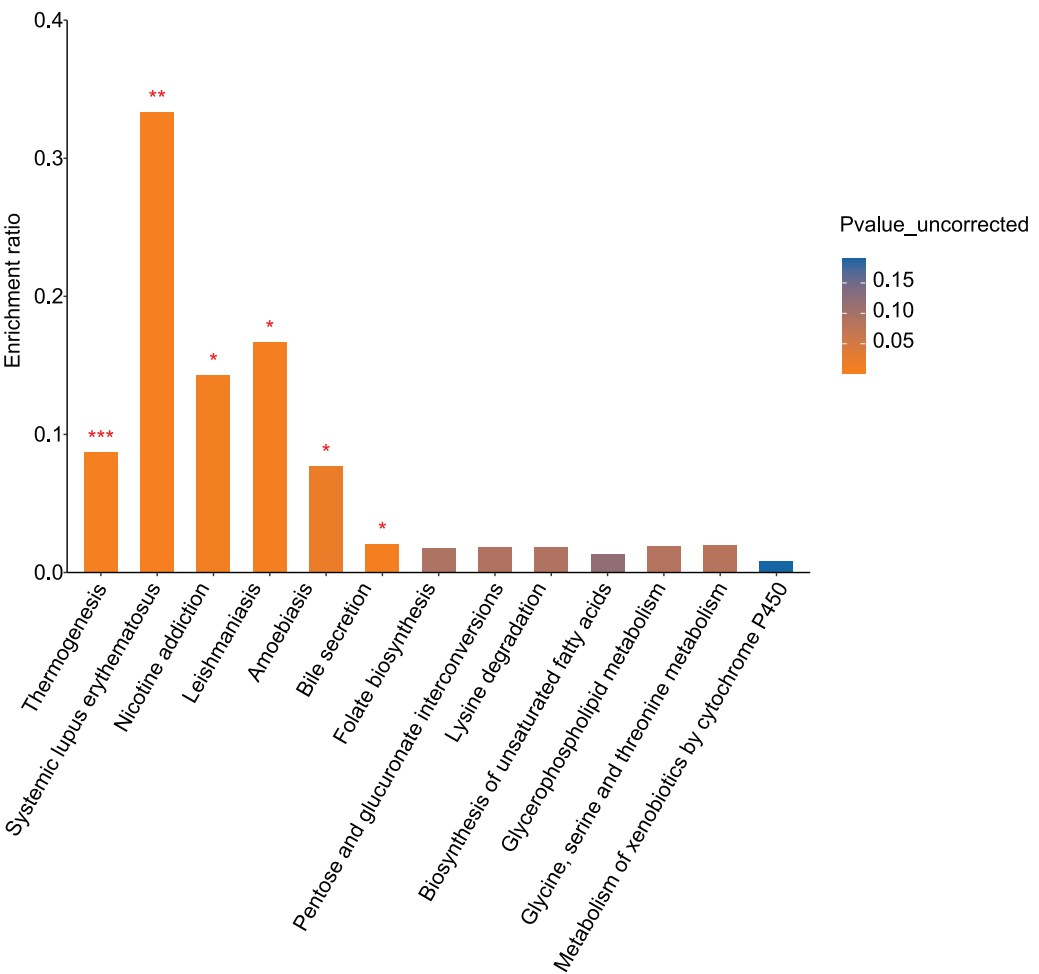

**Figure 4 Pathway analysis and enrichment of differential metabolites between the control and alcohol groups (\*$p < 0.05$, \*\*$p < 0.01$, or \*\*\*$p < 0.001$ *vs* control).**

results show that this model also has good interpretive and predictive performance. Thus, this statistical approach was used to screen for differential metabolites.

### Statistical analysis of differential metabolites

Comparison of the fasudil and alcohol groups identified 35 differential metabolites ($p < 0.05$, VIP < 1, FC > 1; Fig. 6). Eight metabolites were upregulated and 27 metabolites were downregulated after treatment with fasudil. Comparison with the HMDB database classified differential metabolites into six groups (Fig. 7): Lipids and lipid-like molecules (29.63%); Organoheterocyclic compounds (25.93%); Organic oxygen compounds (14.81%); Organic acids and derivatives (11.11%); Phenylpropanoids and polyketides (11.11%); and nucleosides, nucleotides, and analogues (7.41%). Among them, lipids and lipid-like molecules accounted for the largest proportion of metabolites, including choline, glycerophosphate, 2-octenedioic acid, PE(18:3(6Z,9Z,12Z)/P-16:0), 16-hydroxy hexadecanoic acid, PS(now (z) 9/0-0), lysoPC(15:0), PI(18:0/20:4 (5 z, 8 z, z, z) of 14), and

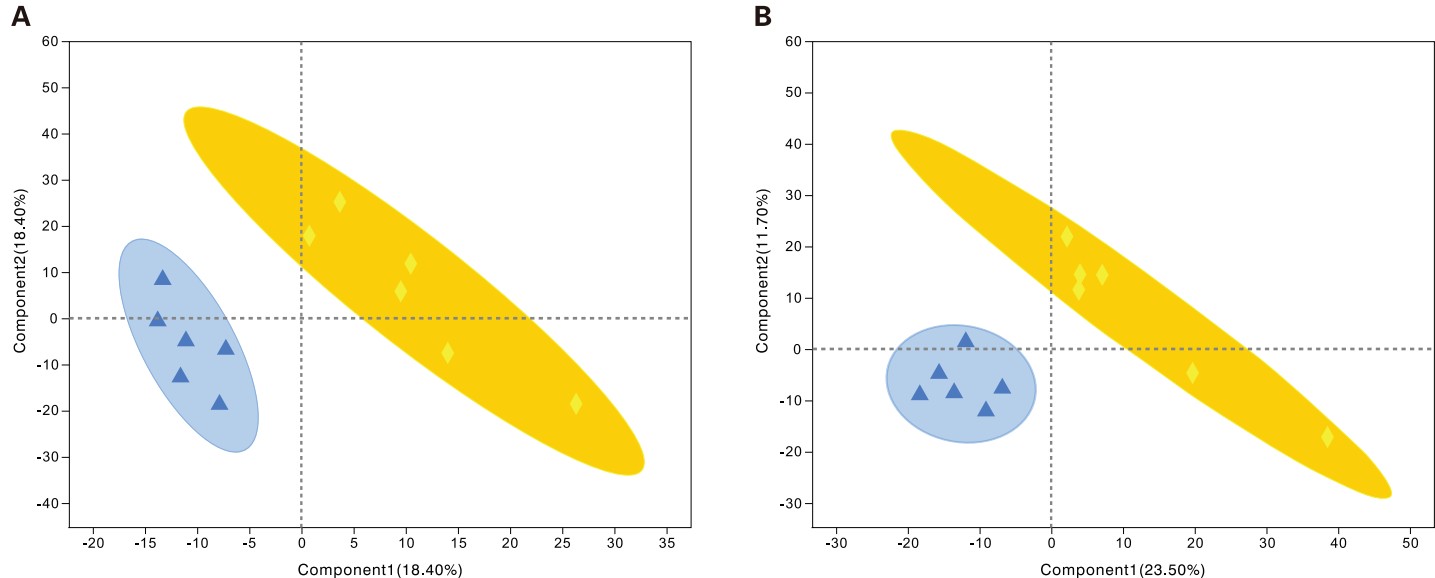

**Figure 5** **Projections to latent structures discriminant analysis (PLS-DA) score plots of astrocytes treated with different media.** Alcohol: normal medium containing 75 mmol/L alcohol, fasudil: normal medium containing 15 μg/mL fasudil. (A) PLS-DA model in the cationic mode. (B) PLS-DA model in the anionic mode. Δalcohol, ♦fasudil.

**Figure 6** **Heatmap of differential metabolites between the alcohol and fasudil groups.** Yellow indicates a greater between-group difference, and blue indicates a lower between-group difference (*p < 0.05, **p < 0.01 vs fasudil).

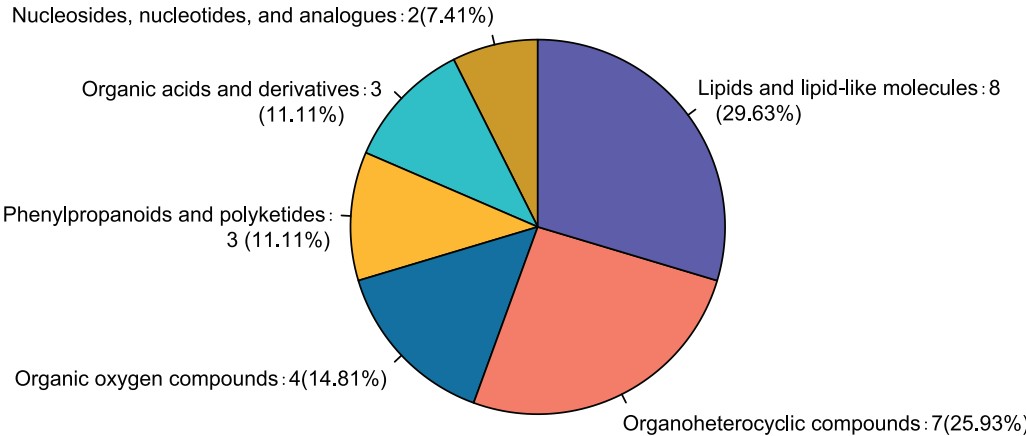

**Figure 7 Classification of differential metabolites and their proportion between the alcohol and fasudil groups.**

lysoPC(18:0). These results suggest that fasudil treatment affected lipid metabolism in astrocytes.

KEGG pathway analysis identified 16 pathways enriched for differential metabolites between the fasudil and alcohol groups. Ether lipid metabolism exhibited the most significant difference ($p = 0$, Fig. 8), implicating the ether lipid metabolism pathway in fasudil-mediated protection of astrocytes after exposure to alcohol.

### Analysis of common differential metabolic pathways

Comparison of metabolomics data from the fasudil and alcohol groups showed that lipids and lipid-like molecules were the primary differential metabolites between the two groups. Furthermore, the KEGG pathway enrichment results showed that the two groups shared three pathways: glycerophospholipid metabolism, bile secretion, and folate biosynthesis. Collectively, these results suggest that lipid metabolism plays important roles in the response to alcohol and the effect of fasudil in astrocytes.

## DISCUSSION

Alcohol may directly or indirectly affect peripheral nicotinic acetylcholine receptor (nAChR) function (*Forman, Righi & Miller, 1989*) and may directly or indirectly activate the reward system in the brain (*Steensland et al., 2007*). Nicotine inhibits astrocyte apoptosis by stimulating α7-nAChRs (*Liu et al., 2015*), and nAChR activation inhibits inflammation in astrocytes (*Kume & Takada-Takatori, 2018*). In this study, metabolite levels related to the KEGG "Nicotine addiction" pathway and "L-nicotine" were significantly different between the alcohol and control groups. Specifically, metabolites related to the "Nicotine addiction" pathway were decreased in the alcohol group. In addition, the amount of L-nicotine present in rat CTX TNA2 astrocytes was significantly decreased after exposure to alcohol. Collectively, these results suggest that alcohol mediates astrocyte function and quantity by regulating molecular components of the "Nicotine addiction" pathway and nAChRs, identifying astrocyte metabolism as a potential therapeutic target for alcohol withdrawal.

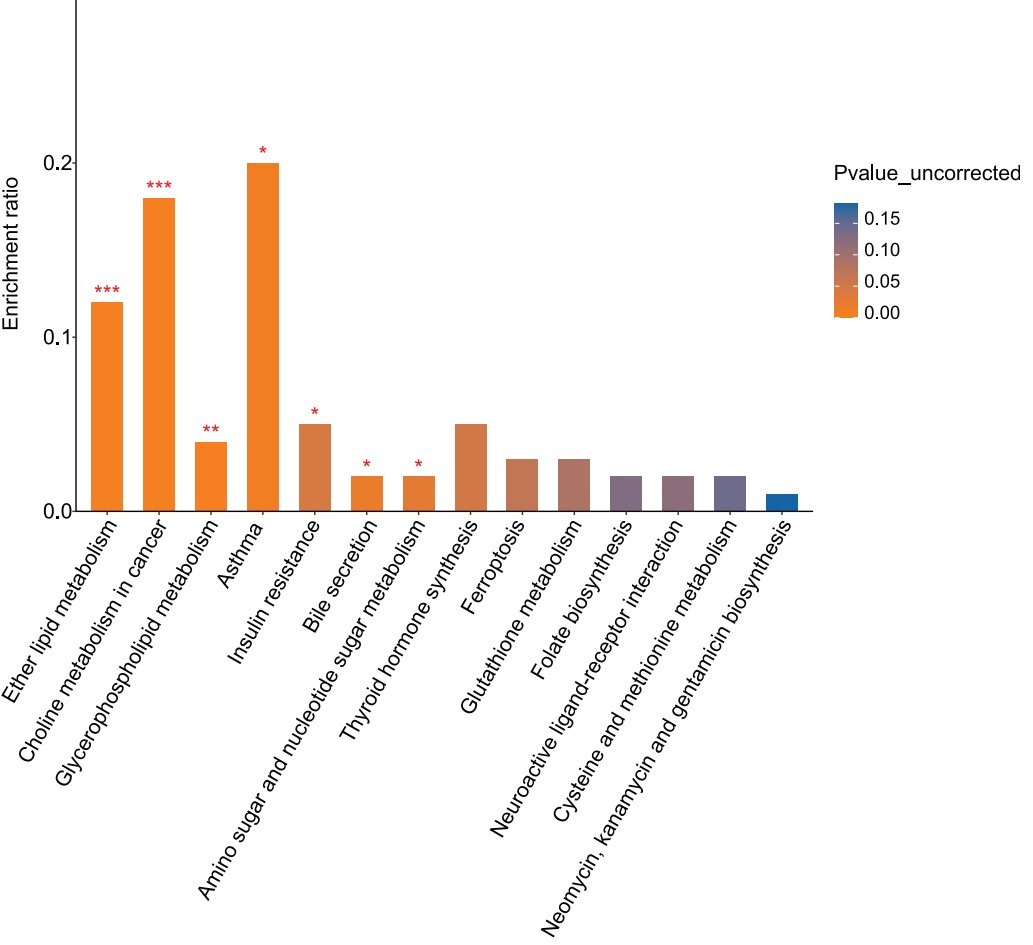

**Figure 8 Pathway analysis and enrichment of differential metabolites between the alcohol and fasudil groups (*$p < 0.05$, **$p < 0.01$, or ***$p < 0.001$ *vs* fasudil).**

The greatest proportion of differential metabolites between the control and alcohol groups consisted of lipids and lipid-like molecules, including various fatty acids and conjugates, as well as glycerophospholipids. Fatty acids are the precursors of most types of lipids that can be synthesized and stored in the cytoplasm. Thus, changes in fatty acid concentrations may alter levels of other lipids. Considerable evidence shows that alcohol primarily affects membrane structures (*Hungund & Mahadik, 1993*). Numerous lipids, such as phospholipids, cholesterol, and ether lipids, play significant roles in regulating cell structure and function, including the formation and function of cell membranes and neurites (*Lee et al., 2021*). In addition, lipids are essential for the functions of astrocytes in the CNS. First, glycogen and lipids are synthesized and released by astrocytes as a form of energy storage, thus providing nutritional metabolic support to neurons (*Corbit et al., 2003*). Second, phospholipids can affect the differentiation of neural stem cells into astrocytes through MAPK/ERK signaling (*Corbit et al., 2003*; *Montaner et al., 2018*). Third, various lipids, including free fatty acids, are involved in the inflammatory regulation of astrocytes. For example, saturated fatty acids can induce TNF-α and IL-6 secretion by

astrocytes, while palmitic acid can mediate cytokine release through TLR4 (*Gupta et al., 2012*). Alcohol may also cause tissue damage through lipid peroxidation (*Li et al., 2020b*). Fourth, a variety of lipid transmitters have been implicated in the addiction process in recent years. For example, the endocannabinoid system is a lipid neurotransmitter complex involved in both development and drug addiction. Oleoylethanolamide (OEA), an endogenous acylethanolamide, is a component of the endocannabinoid lipid system that plays anti-inflammatory, antioxidant, and neuroprotective roles in alcohol-mediated injury (*Antón et al., 2017*). OEA preconditioning inhibited alcohol-induced inflammation, prevented alcohol-induced lipid peroxidation and nerve damage, and reduced alcohol intake (*Tellez et al., 2013*). Although the exact mechanism is unclear, Tellez et al. suggested that the effects of OEA may be mediated by the brain's reward system (*Forgione & Fehlings, 2014*). In addition, numerous experiments have successfully confirmed that lipids can influence the activation and viability of astrocytes and neurons (*White, Ellis & Wolfgang, 2021*; *Shano et al., 2008*; *Song et al., 2021*; *Misslin et al., 2017*). Thus, we conclude that there is an inevitable connection among ethyl alcohol, lipids, astrocytes, and neurons.

Our findings identify metabolic differences of certain lipids, such as 11,14,17-eicosatrienoic acid, tetradecanedioic acid, palmitoyl glucuronide, PS(18:0/18:1(9Z)), and PG(18:1(11Z)/22:6(4Z,7Z,10Z,13Z,16Z,19Z)). Presumably, these lipids may act as mediators through which ethyl alcohol affects astrocytes and even the entire CNS.

Rho kinases (ROCK1 and ROCK2) are serine/threonine protein kinases that regulate cell migration, proliferation, and survival, as well as axon growth and dendrite endocytosis (*Julian & Olson, 2014*). The ROCK pathway has been implicated in the pathogenesis of many common neurodegenerative diseases. Moreover, beneficial neurological effects of Rho kinase inhibitors have been observed in animal models of nervous system injury (*Bito et al., 2000*; *Zhou et al., 2003*; *Eldawoody et al., 2010*). In 2015, *Kurt et al. (2015)* first reported an association between Rho kinase activity and alcohol intake and withdrawal in rats, confirming that drugs affecting Rho kinase could help treat alcohol withdrawal syndrome. ROCK2 inhibition also has reported anti-apoptotic effects and can improve alcohol-mediated neuronal damage, possibly by inhibiting the nuclear factor κB pathway (*Li et al., 2020a*). Fasudil, an effective small-molecule inhibitor of ROCKs, has been used to treat neurodegenerative diseases. Therefore, in this study, fasudil was used to treat astrocytes that had undergone alcohol-induced injury, and the cells were subjected to metabolomic analysis. The most striking differential metabolites between the control and alcohol groups were lipids and lipid-like molecules, as well as the ether lipid metabolism pathway. As mentioned above, ether lipid is a basic CNS component. AD is a chronic disease involving CNS injury caused by long-term alcohol intake. Our results suggest that fasudil can promote axonal regeneration, reduce inflammation, and decrease oxidative stress in alcohol-injured neurons by affecting lipid metabolism through a variety of pathways, such as ether lipid metabolism.

Previous studies demonstrated wide involvement of the ROCK pathway in lipid metabolism and a positive role in dyslipidemia models (*Ma et al., 2011*; *Li et al., 2015*). Although fatty acid synthesis and oxidation are controlled by peroxisome proliferator-activated receptor (PPAR)-α and steroid-response element-binding protein 1

(SREBP-1), alcohol also inhibits adenosine monophosphate-dependent protein kinase (AMPK) to control fatty acid metabolism. ROCK1 in peripheral tissues also inhibits AMPK2a and indirectly inhibits SREBP-1, ultimately reducing energy expenditure and increasing fat synthesis (*Landry, Shookster & Huang, 2021*). Knockdown of ROCK1 expression in healthy mice led not only to excessive food intake, dyslipidemia, and obesity, but also hyperleptinemia (leptin is an effective adipokine) (*Landry, Shookster & Huang, 2021*). Leptin levels can increase with increasing chronic alcohol intake (*Kiefer et al., 2001*), indicating that ROCK and alcohol may affect lipid metabolism through many of the same pathways. Notably, although fasudil can also inhibit other Rho kinase-related proteins, such as protein kinases A (PKA) and G (PKG), the degree of inhibition is mild. A large amount of literature confirms that PKA/PKG/PKC can affect lipid metabolism (*Turchi et al., 2020*; *Keenan et al., 2021*; *Liang et al., 2020*; *Rabhi et al., 2018*; *Yang et al., 2019*; *Peng et al., 2015*; *Huang et al., 2014*; *Ranganathan et al., 2002*) and reduce the damage caused by oxidative stress (*Kato et al., 1991*). PKA plays a vital role in regulation of lipid metabolism, which allows it to serve as a key factor in cell signaling. PKA activates lipases, including hormone-sensitive lipase and fatty triglyceride lipase, to promote fat mobilization. *Huang et al. (2017)* showed that PKC inhibitors could restore glutamate transporter one levels on the surface of astrocytes treated with PPAR-α agonists to regulate astrocytic lipid metabolism and functions. Thus, our experiments confirm that fasudil can influence the lipid metabolism of astrocytes by inhibiting the ROCK pathway or influencing PKA, PRG, PKC, and/or related proteins; however, further experiments are needed to confirm this conjecture.

In addition, our study had some shortcomings. CTX TNA2 cells (type 1 astrocytes) were unable to completely replicate astrocytes, let alone a variety of human astrocytes, *in vitro*. Henceforth, we will conduct additional experiments to acquire sufficient evidence, and further experiments are needed to investigate the relationships between other metabolic pathways, astrocyte secretion, and AD.

## CONCLUSIONS

Our experimental results showed that lipids and lipid-like molecules were differentially present in the alcohol and fasudil groups, which shared glycerophospholipid metabolism as an enriched metabolic pathway. These results suggest that lipids are an important target of astrocyte and/or glia-neuron function regulation in AD. Both alcohol and fasudil can disrupt intracellular cholesterol homeostasis and affect cell function. However, fasudil may prevent and even repair astrocytes by affecting other lipid metabolism pathways, especially the glycerophospholipid and ether lipid pathways. Our findings show that fasudil may help prevent alcohol-induced astrocyte damage by modifying lipid metabolism, suggesting that it is a candidate for treatment of AD.

### Funding

Xunzhong Qi received funding from the Doctoral Initiation Project of Jiamusi University (grant numbers: JMSUBZ2019-06) and the Talent training project for basic scientific research of Heilongjiang Province Educational Commission of China (grant numbers: 2019-KYYWF-1357). Xiaofeng Zhu received funding from The National Key R&D Program of China (grant numbers: 2018YFC1314404), the Shunjie Bai received funding from the National Key R&D Program of China (grant numbers: 2018YFC1314404); National Natural Science Foundation of China (grant numbers:81901398), and the Natural Science Foundation of Chongqing, China (grant numbers: cstc2019jcyj-msxmX0025). The funders had no role in study design, data collection and analysis, decision to publish, or preparation of the manuscript.

### Grant Disclosures

The following grant information was disclosed by the authors:
Doctoral Initiation Project of Jiamusi University: JMSUBZ2019-06.
Heilongjiang Province Educational Commission of China: 2019-KYYWF-1357.
National Key R&D Program of China: 2018YFC1314404.
National Key R&D Program of China: 2018YFC1314404.
National Natural Science Foundation of China: 81901398.
Natural Science Foundation of Chongqing, China: cstc2019jcyj-msxmX0025.

### Competing Interests

The authors declare that they have no competing interests.

### Author Contributions

- Huiying Zhao performed the experiments, prepared figures and/or tables, and approved the final draft.
- Xintong Li performed the experiments, prepared figures and/or tables, and approved the final draft.
- Yongqi Zheng performed the experiments, prepared figures and/or tables, and approved the final draft.
- Xiaofeng Zhu conceived and designed the experiments, authored or reviewed drafts of the article, and approved the final draft.
- Xunzhong Qi conceived and designed the experiments, prepared figures and/or tables, and approved the final draft.
- Xinyan Huang performed the experiments, authored or reviewed drafts of the article, and approved the final draft.
- Shunjie Bai analyzed the data, authored or reviewed drafts of the article, and approved the final draft.
- Chengji Wu analyzed the data, authored or reviewed drafts of the article, and approved the final draft.

- Guangtao Sun conceived and designed the experiments, analyzed the data, authored or reviewed drafts of the article, and approved the final draft.

## Data Availability

The data is available at MetaboLights: MTBLS5071.

https://www.ebi.ac.uk/metabolights/editor/MTBLS5071/.

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
