# Peer review of "Fasudil may alleviate alcohol-induced astrocyte damage by modifying lipid metabolism, as determined by metabonomics analysis"

_PeerJ, doi:10.7717/peerj.15494_

## Round 0.1 · original submission · Minor Revisions

This study investigated alcohol-induced changes in metabolite levels in an astrocyte cell line and the effects of fasudil on these alcohol-induced metabolite changes. The writing of the article needs to be thoroughly reviewed.

Reviewer 1 ·

Basic reporting

This manuscript by Zhao and coworkers titled ‘Fasudil prevents alcohol-induced astrocyte damage by modifying lipid metabolism, as determined by metabonomics analysis’ describes metabolic profiles of astrocytes exposed to alcohol wit and without fasudil compared and compared them to untreated astrocytes. This is an interesting study. The manuscript is written in a clear and unambiguous language. Introduction and discussion were appropriately written. Methods were described adequately clear. Results were clearly written and supported by good quality figures.
I have three important concerns which required to be addressed necessarily in revision.
1. The authors used uncorrected p values in pathway enrichment. Please use p value correction for multiple hypothesis testing.
2. Please show in a table or in figure what metabolites are causing enrichment of each pathway (intersections) shown in figure 5 and 10.
3. Pathway enrichment in metabolomics data often throw false positive enrichment predictions. Please identify what metabolites caused enrichment of lipid metabolism and validate the changes ion concentration using invitro experiment. This is essential to prove that what authors identified in bioinformatic analysis is indeed true without which this study may have serious problems with scientific rigor and validity.

Experimental design

Please refer to basic reporting section.

Validity of the findings

Please refer to basic reporting section.

Reviewer 2 ·

Basic reporting

This study reports changes in astrocyte metabolome following alcohol treatment and fasudil + alcohol treatment. This study is exciting. The investigators used mass spec-based metabolome analysis for delineating molecular mechanism of action of fasudil. This is a well written manuscript except for few overstatements.

Addressing the following issues may help the investigators in making this manuscript better.

1. This manuscript did not have any data to confirm that lipid metabolism is indeed responsible for preventing astrocyte damage. This requires sound experimental evidence. However, the only evidence in this manuscript in this regard is bioinformatic analysis-based predictions. Therefore, I request authors to modify the title of the manuscript tone it down. Similarly, tone down the sentences in abstract, results and discussion.
2. How many metabolites were detected in mass spec? How were missing values handled?
3. How many metabolites were enriched in each pathway term? Please show p.adj for all pathways?
4. Did you use fold change cut off for metabolites used as input in pathway analysis?
5. Figure legends are less informative. I request authors to elaborate legends.
6. Authors may reduce total figures numbers by incorporating multiple figure panels in one figure.
7. Please make sure to choose colors for figures to improve visibility (Figure 6&7).
8. Please label color keys in figure 3. What p values ranges are indicated by asterisks shown in this figure?

Experimental design

Please see above.

Validity of the findings

Please see above.

·

Basic reporting

There are too many references. If the number is reduced and adding some more recent references will be better.

Experimental design

no comment

Validity of the findings

no comment

Additional comments

The study methodology is described in detail. The review of the literature seems adequate and appropriate references have been provided. The results have been discussed in an unbiased manner.

However, the manuscript needs some editing by a fluent English speaker to improve the grammar/syntax.

·

Basic reporting

In this study, the authors investigated about the changes of alcohol-induced metabolite levels in an astrocyte cell line, and the effects of fasudil on that alcohol-induced metabolite changes. The paper is poorly written and needs to be amended extensively. My comments are given below:
Introduction:
The authors should state the background and purpose of the study here. Since, they checked the effects of alcohol treatment on astrocytes, and they modeled it as human alcohol dependency, they need to describe details about changes of astrocytes in alcohol dependency in human, and what previous research showed in this matter, and what is not known. Then, what is the importance of Rho kinases in alcohol dependency. Also, why they checked astrocyte-derived metabolites, needs to be stated in the introduction. They said in the introduction that alcohol dependency is a learning and memory disorder. Is it true? Overall, the introduction should be totally rewritten.
Methods:
The authors used 5 mmol/L concentration of alcohol and 15 micrograms/ml of fasudil. What is the rationale to use these specific concentrations, the authors need to state in the introduction. Is the concentration of alcohol mimics the concentration of during alcohol dependency? This point needs to be stated in the introduction.
Cell culture methods and cell treatment: The authors need to say clearly about the method of cell culture and treatment, the composition of cell culture medium, medium used during treatment. They said during treatment they used complete medium. But the term ‘complete medium’ is used to describe the medium used to culture the cells, not treatment medium.
Metabolite extraction method needs to be described clearly. In the LS-MS, mobile phase B contains 49.5% acetonitrile + 49.5% ethylpropanol + 0.1% formic acid. Is the rest 0.9% water? Please describe the method clearly.
I do not understand what is the quality control’ sample, and how it ensure the stability and repeatability of the experiment.

Results:
After reading, I do not understand much about the results. PLS-DA was used to separate the groups. The authors should state which metabolites are important to discriminate by PLS-DA. Probably, the tried to say that using heat map, but it is difficult to understand.
The authors should reduce the figure number by combining them. In this way the result might be presented in a better way.
Discussion:
The discussion should be based on the results, and should not be out of line. For example, the authors discussed about TLR4 and other signaling. Is it OK?
In pathway analysis, bile secretion, Amoebiasis, Leishmaniasis pathways were enriched. What is that mean? Is there any relation of bile secretion pathway and any of astrocyte function?
Language: The language should be improved. Sometimes a word is used repeatedly, which is not very attractive to read. For example, in the Materials section, the word ‘purchase’ is repeatedly used.
What is ‘L-metabolism’.

Experimental design

The experimental design is not adequately explained in the manuscript

Validity of the findings

The results are poorly written. So, it is difficult to check the validity of the findings

Additional comments

The language quality is poor. The manuscript should be thoroughly checked by a Scientific editor.

---

## Round 0.2 · accepted · Accept

The authors addressed almost all concerns according to the reviewers' comments. Therefore the manuscript can be accepted.

Reviewer 1 ·

Basic reporting

My comments were addressed.

Experimental design

No comments

Validity of the findings

No comments

Reviewer 2 ·

Basic reporting

The authors addressed my concerns satisfactorily. I recommend this manuscript for publication.

Experimental design

.

Validity of the findings

.

·

Basic reporting

No comment

Experimental design

No comment

Validity of the findings

No comment

Additional comments

No comment